# Experimental and Numerical Study of Slug-Flow Velocity Inside Microchannels Through In Situ Optical Monitoring

**DOI:** 10.3390/mi16050586

**Published:** 2025-05-17

**Authors:** Samuele Moscato, Emanuela Cutuli, Massimo Camarda, Maide Bucolo

**Affiliations:** 1Department of Electrical Electronic and Computer Science Engineering, University of Catania, Via Santa Sofia 64, 95125 Catania, Italy; emanuela.cutuli@phd.unict.it (E.C.); maide.bucolo@unict.it (M.B.); 2STLab srl, Via Anapo 53, 95126 Catania, Italy; massimo.camarda@stlab.eu; 3Dipartimento di Fisica e Astronomia ‘Ettore Majorana’, Università degli Studi di Catania, Via Santa Sofia 64, 95123 Catania, Italy; 4Istituto per la Microelettronica e Microsistemi CNR-IMM, Sezione di Catania, Strada VIII Zona Industriale 5, 95121 Catania, Italy

**Keywords:** two-phase flow, microfluidics, micro-optics, computational fluid dynamics, experimental validation

## Abstract

Miniaturization and reliable, real-time, non-invasive monitoring are essential for investigating microfluidic processes in Lab-on-a-Chip (LoC) systems. Progress in this field is driven by three complementary approaches: analytical modeling, computational fluid dynamics (CFD) simulations, and experimental validation techniques. In this study, we present an on-chip experimental method for estimating the slug-flow velocity in microchannels through in situ optical monitoring. Slug flow involving two immiscible fluids was investigated under both liquid–liquid and gas–liquid conditions via an extensive experimental campaign. The measured velocities were used to determine the slug length and key dimensionless parameters, including the Reynolds number and Capillary number. A comparison with analytical models and CFD simulations revealed significant discrepancies, particularly in gas–liquid flows. These differences are mainly attributed to factors such as gas compressibility, pressure fluctuations, the presence of a liquid film, and leakage flows, all of which substantially affect flow dynamics. Notably, the percentage error in liquid–liquid flows was lower than that in gas–liquid flows, largely due to the incompressibility assumption inherent in the model. The high-frequency monitoring capability of the proposed method enables in situ mapping of evolving multiphase structures, offering valuable insights into slug-flow dynamics and transient phenomena that are often difficult to capture using conventional measurement techniques.

## 1. Introduction

Microfluidics is a rapidly growing field that involves the precise manipulation of small volumes of fluids at the microscale for applications in biotechnology [1,2], medicine [3,4], and chemical [5] and biochemical analysis [6]. In these microscale environments, fluid dynamics is governed by low Reynolds number regimes, where viscous and surface tension forces dominate over inertial effects, resulting in highly stable and predictable laminar flow [7]. A thorough understanding of fluid behavior in microfluidic systems is essential for optimizing device designs, enhancing performance, and ensuring reliability in real-world applications [8,9]. To achieve this, progress in the field relies on three primary approaches: analytical modeling, computational fluid dynamics (CFD) simulations, and experimental validation methods, each offering unique strengths and limitations, as outlined in the SWOT analysis presented in Appendix A.

Analytical models provide mathematical solutions based on fundamental fluid dynamics principles, offering quick insights into simple geometries and flow conditions [10]. However, their applicability is often limited in complex microfluidic systems involving nonlinear effects, non-Newtonian fluids, or intricate geometries [11]. CFD simulations address some of these challenges by numerically solving the governing equations of fluid dynamics, enabling a detailed analysis of flow behavior in microfluidic designs [12]. Despite their advantages, CFD models require significant computational resources and validation against experimental data [13,14]. In this context, experimental methods serve as essential validation tools, providing real-world high-accuracy measurements of fluid behavior within microfabricated devices [15]. One of the key advantages of experimental models is their ability to incorporate physical phenomena that are often oversimplified or neglected in computational models. Many microfluidic systems exhibit nonlinear effects, such as viscoelastic behavior [16], multiphase interactions [17], and inertial instabilities [18], which can be challenging to accurately model and simulate. For instance, fluids with complex rheological properties, such as non-Newtonian fluids or polymer solutions, display shear-thinning or shear-thickening behaviors that analytical and CFD models often approximate using empirical constitutive equations [19]. Furthermore, CFD simulations frequently assume static or quasi-static conditions, making it difficult to capture transient behaviors that are crucial in microfluidics [20]. Many CFD models also rely on idealized wetting conditions, assuming constant contact angles or uniform surface properties, which may not fully reflect real-world scenarios [21,22,23].

In the continuously expanding field of microfluidic research, many real-world applications involve dynamic processes, such as droplet formation, mixing, and transient flow instabilities, where time-dependent changes significantly affect system performance [24]. Slug-flow processes, in particular, are extensively studied in microfluidics due to their critical role in multiphase flow applications, including heat transfer and micromixing systems [25], droplet-based microfluidics [26], Lab-on-a-Chip (LoC) systems [27], and microreactors [28]. This flow regime, characterized by alternating liquid segments (slugs) separated by gas or immiscible liquid phases, exhibits complex interfacial dynamics, transient behavior, and nonlinear effects that greatly influence overall system performance [29]. Slug-flow processes often involve phase interactions, such as recirculating vortices within liquid slugs [30] or the presence of a meniscus [31], which are typically considered secondary effects or simplified in multiphase CFD models due to computational limitations.

Accurate and reliable estimation of slug-flow velocity is crucial, as it provides experimentally validated, real-world data that analytical models and CFD simulations often fail to capture due to the complexity of the real fluid dynamics model and high computational times, directly impacting key microfluidic applications. In slug-based microreactors, for example, the internal circulation within liquid slugs enhances mixing, which is vital for efficient chemical reactions. Inaccurate velocity estimation can lead to under-mixing or over-mixing, negatively affecting reaction yields and product uniformity [32]. Similarly, in microscale heat exchangers or biochemical assays, slug velocity determines the fluid residence time, influencing mass and heat transfer rates [33]. In droplet generation systems, velocity governs droplet formation, size uniformity, and stability [34]. Additionally, in biomedical and cell-based applications, velocity directly affects shear stress, nutrient transport, and cell viability [35], ensuring optimal conditions for processes such as cell culture [36], drug delivery [37], and organ-on-a-chip systems [38]. Given the importance of these applications, integrating experimental data with computational models is crucial for achieving a complete understanding of microscale fluid behavior.

This study aims to expand on the outcomes of previous experimental works [39,40]. The novelty of this research lies in the development of an on-chip experimental approach to estimate slug-flow velocity within microchannels using in situ optical monitoring in a simple and reliable manner. By validating and comparing the collected experimental results with analytical models and CFD simulations, this holistic strategy enables a more detailed understanding of slug-flow dynamics inside microchannels. For this purpose, we employed a Polydimethylsiloxane (PDMS)-based micro-optofluidic (MoF) device, previously detailed in [39,40], featuring a T-junction microfluidic geometry for slug-flow formation and integrating micro-optical components, including a fully integrated micro-optical light splitter coupled with optical fibers, which enables label-free velocity estimation via optical signals.

Recent advances in micro-optofluidics have demonstrated the successful integration of optical fibers into microfluidic platforms [41,42,43] due to their compact form factor, immunity to electromagnetic interference, low sample consumption, high sensitivity, and ability to provide dynamic, real-time measurements [44]. For example, in [45], a microflow cytometer employing viscoelastic focusing was presented, utilizing optical detection via fiber-coupled light sources and photodetectors. In [46], an optofluidic fiber sensor was developed for real-time refractive index monitoring. Another study [47] proposed a polymer-based Mach–Zehnder interferometer (MZI) printed directly onto a single-mode fiber. In [48], D-shaped fiber Bragg gratings (FBGs) were used to monitor two-phase flows in microchannels, enabling the quantification of droplet velocity and size. While these approaches provide excellent sensitivity and resolution, they often require complex alignment and immobilization of optical fibers, involve sophisticated fabrication steps, or lack full integration with microfluidic components. In contrast, the platform proposed in this work introduces a fully integrated optofluidic design, in which the optical waveguide is directly bonded to the PDMS microfluidic layer. This configuration significantly simplifies the assembly process while maintaining high-precision measurement capabilities. Another key advantage of the proposed approach lies in its reliance on optical signal acquisition, which enables high-frequency monitoring of slug-flow dynamics. This allows for precise and non-invasive mapping of the evolving multiphase structures and their interactions within the microchannel. The ability to acquire high-resolution real-time data provides valuable insight into transient phenomena that are typically difficult to capture with conventional measurement techniques.

This study investigates the limitations and advantages of numerical predictions compared to experimental observations in characterizing slug formation and laminar-flow regimes. Notable discrepancies between predicted and real process conditions, especially in gas–liquid flows such as air–water slug regimes, highlight the limitations of purely theoretical models. These deviations underscore the critical need for in situ real-time flow monitoring to ensure accuracy in applications where precise control is essential.

This paper is organized as follows. Section 2 discusses the materials and methodologies, including the design of the MoF device (Section 2.1) and the CFD implementation (Section 2.2). Section 2.3 and Section 2.4 outline the experimental setup and campaign. The results of the liquid–liquid and gas–liquid slug flows are presented in Section 3, comparing the analytical, simulation, and experimental data. The conclusions and future developments are presented in Section 4.

## 2. Materials and Methods

### 2.1. Device Design and Working Principle

The geometry of the micro-optofluidic (MoF) device employed in this work was designed through appropriate simulations, and further details have been provided in previous studies [39,40]. Briefly, the device features two interconnected microchannels forming a T-junction for the slug-flow formation. It includes two inlets for introducing the fluids into the microfluidic channel and a single outlet for fluid discharge. The optical component of the MoF device consists of a micro-optical light splitter, used as a waveguide to reorient a light beam, along with three optical fiber insertions, schematized in the figure by gray-colored cylinders. The input optical fiber (see Figure 1, upper-right corner) conveys the light source to the microsplitter, which provides as output a cone-shaped beam of light with a direction angle of about $36^{\circ }$ [39] because of its numerical aperture toward the microchannel’s investigation region. As the light interacts with the two fluids that compose the slug flow, the optical signals are captured by two output optical fibers arranged on the opposite side of the microchannel (see Figure 1, bottom part). The two output optical fibers are positioned at an angle of $36^{\circ }$ relative to the vertical direction in order to align with the light beam coming from the light source. The details regarding the design optimization process through optical simulation are provided in [39], and the precise dimensions of the MoF device are reported in Figure 2a.

The MoF device is designed to detect immiscible gas–liquid and liquid–liquid slug flows and evaluate their velocities using optical signal-based methods. Its working principle is illustrated in Figure 1 and relies on the absorption phenomenon. Depending on the fluid’s refractive index value, the interaction with the incident laser beam determines different light transmission. Consequently, the acquired optical signal exhibits different amplitudes depending on the fluid with which it is interacting at a given instant. More specifically, the acquired optical signal has a square-wave shape, characterized by two levels corresponding to each fluid making up the slug flow [39].

#### Device Manufacturing

The MoF device was fabricated using PDMS with a master–slave approach based on inkjet 3D-printing technology [49]. This method, which falls within the category of soft lithography manufacturing methods, overcomes the challenges associated with traditional microfabrication techniques, like photolithography, including high costs, complex processing, clean-room requirements, expensive equipment, and hazardous chemical handling.

The fabrication process began with the mold design (see Figure 2a) in Autodesk^®^ Fusion 360 (v.2.0.17721) software, followed by slicing and G-code generation using Objet Studio^TM^ software (v.9.2.11.6825, Stratasys, Los Angeles, CA, USA). The mold was then fabricated using a PolyJet 3D printer Stratasys Objet260 Connex 1 (Stratasys, Los Angeles, CA, USA), which deposits and UV-cures liquid photopolymer ink (Vero PureWhite RDG837, OVERMACH S.p.A, Parma, Italy) layer by layer. Post-processing involved washing the support material (FullCure705, OVERMACH S.p.A, Parma, Italy) away and an additional UV treatment at 35 °C for 1 h to ensure surface quality. The final 3D-printed mold is shown in Figure 2b.

The PDMS mixture was prepared by mixing a silicone elastomer base with a curing agent (Sylgard 184 elastomer kit, Dow Corning, Midland, MI, USA) at specific ratios of 10:1 for the device layer and 5:1 for the bulk cover layer, followed by degassing under vacuum to remove bubbles generated during mixing. The mixture was poured into the 3D-printed mold, cured at room temperature for 48 h, and then demolded and bonded to a bulk cover 0.5 mm thick using a reversible bonding procedure. To enhance adhesion, both layers were treated with plasma, using Elveflow’s Plasma Bonding Pen (Elveflow, Paris, France) for 3 min each. The previously 3D-printed micro-optical light splitter [39] was inserted into its designated slot, and the bonding process was completed with a thermal treatment at 80 °C on a heating plate for 15 min. The final assembled MoF device is shown in Figure 2c. For more details about the device manufacturing, refer to Appendix A.

### 2.2. Slug-Flow Computational Model

#### 2.2.1. Theoretical Background

Slug flow is a specific regime of two-phase flow and refers to a condition in which two different immiscible fluids flow intermittently as alternating phases of one fluid (slugs) separated by the other. One of the most effective methods for modeling two-phase processes is to integrate the Navier–Stokes (NS) equations with the phase-field (PF) method [50], which describes the principles of momentum conservation and mass conservation for Newtonian fluids using the continuity equation. Specifically, the equations that govern the transition between two immiscible and incompressible phases play a crucial role in this modeling process and are described as follows:(1)$$\rho \left(\frac{\partial v}{\partial t}+v\cdot \nabla v\right)=\rho g-\nabla p+\nabla \mu \left(\nabla v+{\left(\nabla v\right)}^{⊤}\right)+G\nabla \phi$$(2)∇·v=0
where ρ is the density expressed in [kg/m^3^]; *v* is the flow velocity expressed in [m/s]; g is the acceleration induced by external forces such as gravitational, magnetic, and electrostatic forces, expressed in [m/s^2^]; p is the pressure measured in [N/m^2^]; μ is the viscosity coefficient expressed in [Pa s]; *G* is the chemical potential, which expresses the rate of change in free energy measured in Gibbs [G]; and ϕ represents the dimensionless phase-field variable, which is used to model the interface dynamics in the phase-field tracking method [51,52]. In particular, the area where ϕ∈(−1,1) corresponds to the fluid interface region, and when ϕ=±1, the area possesses the properties of the specific fluid. Specifically, ϕ=1 refers to the *continuous phase*, while ϕ=−1 refers to the *dispersed phase*. Moreover, ρ and μ are strictly related to ϕ through the relations(3)$$\rho =\frac{1-\phi }{2}\rho_{1}+\frac{1+\phi }{2}\rho_{2}$$(4)$$\mu =\frac{1-\phi }{2}\mu_{1}+\frac{1+\phi }{2}\mu_{2}$$
where $\rho_{1}$ and $\mu_{1}$ are the density and viscosity of the *continuous phase*, and $\rho_{2}$ and $\mu_{2}$ are the density and viscosity of the *dispersed phase*.

This representation, while highly realistic, presents challenges due to its complexity, as well as the computational demands and difficulties associated with modeling boundary conditions between two immiscible fluids. In order to overcome these problems, several dimensionless parameters are employed to characterize different aspects of fluid flow by directly using input fluid flow rates and channel geometry information. Among these parameters, the Reynolds number (Re) and the Capillary number (Ca) offer valuable insights into microfluidic flow dynamics. Specifically, the Reynolds number is utilized to assess the relative significance of inertial forces compared to viscous stresses, thereby facilitating the prediction of flow behavior as laminar or turbulent. Conversely, the Capillary number serves to evaluate the relative importance of surface tension forces in comparison to viscous forces within the fluid flow, aiding in the determination of the flow regime.

#### 2.2.2. Computational Fluid Dynamics (CFD) Model

Computational fluid dynamics (CFD) was used to model the micro-optofluidic device and investigate both the liquid–liquid and gas–liquid slug flows to facilitate a direct comparison between the experimental results and numerical representations. Specifically, the model was implemented using COMSOL Multiphysics software (v5.5). The model was designed to simulate only the microfluidic process and not the optical process, which was previously investigated and modeled in [39]. The model structure—including microchannel dimensions, geometry, meshing, and numerical methods—was implemented consistently for both processes. The only significant difference was in the materials’ domains and their physical properties: one case involved liquid–liquid interaction, while the other addressed gas–liquid interaction. Specifically, for the liquid–liquid slug flow, hexadecane was used as the *continuous phase* and deionized water as the *dispersed phase*, whereas for the gas–liquid slug flow, deionized water was used as the *continuous phase* and air as the *dispersed phase*.

The microchannel geometry consisted of a T-junction with both central and lateral inlet microchannel widths of 400 [μm] and an outlet microchannel width of 400 [μm], as shown in Figure 3a. These dimensions were chosen to replicate the microchannel geometry used, as described in Section 2.1. Taking into account the process characteristics, the simulation time step was set in the range between 1 and 10 [ms], and an unstructured triangular mesh was considered for the space discretization to achieve the best contoured fitting along the T-junction (see Figure 3b). Moreover, in order to establish the same analysis methodologies between the experimental and numerical results, the optical components corresponding to the two photodiodes were simulated using two observation points denoted as $P_{1}$ and $P_{2}$, as shown in Figure 3c. Indeed, these points were positioned at the same locations as the optical fibers to capture their respective patterns.

To establish the model configuration, several physical and numerical parameters needed to be established. The atmospheric pressure and room temperature were set at P=1 [atm] and T=20 °C, respectively. The surface tension coefficients for hexadecane–water and air–water were $s_{t(HW)}=0.051$ [N/m] [53] and $s_{t(AW)}=0.072$ [N/m] [54], respectively. Moreover, according to the existing literature [55], the contact angle at the fluid interface was assigned a value of $\theta_{c}=135^{\circ }$ for both the liquid–liquid and gas–liquid interactions. To define the initial conditions for the fluids, the geometry was divided into two domains: $D_{1}$, which included the central inlet and the main microchannel, and $D_{2}$, which included the lateral microchannel (see Figure 3a). As an initial condition, the domain $D_{1}$ was filled with the *continuous phase*, while the domain $D_{2}$ was filled with the *dispersed phase*. Additionally, it was assumed that the fluids at the wall were incompressible flows and stationary, adhering to a no-slip condition. At the outlet section, the pressure was assumed to be zero, and the initial fluid interface was established at the boundary between $D_{1}$ and $D_{2}$.

The model configuration employed a multiphysics approach by coupling laminar-flow (*LF*) and phase-field (*PF*) physics within a two-dimensional representation. This choice facilitated the precise tracking of the interface between the two fluids considered. Moreover, the adoption of a two-dimensional model was primarily driven by the need to optimize computational resources, as simulating microfluidic processes, especially those involving multiphase flow, can be computationally demanding. Moreover, the choice was further justified by the inherent symmetry of the process, which makes a two-dimensional representation both appropriate and effective. Consequently, 2D simulations were conducted under the assumption that the dynamic behavior remained symmetric along the third dimension, thereby reducing computational time. A more detailed discussion of the CFD model used can be found in [51].

### 2.3. Experimental Setup

The system employed to generate the slug flow through the micro-optofluidic device is schematically represented in Figure 4b. It consists of (i) a hydrodynamic system used to introduce fluid samples into the MoF device; (ii) an optical actuation system comprising a light source delivered to the device via an input optical fiber; (iii) the micro-optofluidic device; (iv) an optical detection system utilizing a photodiode connected to the output optical fibers; (v) a scientific camera for image acquisition; and (vi) a computer equipped with dedicated software for optical data analysis. The experimental setup is shown in Figure 4a.

To create the slug flow within the micro-optofluidic device, two fluids were simultaneously injected through two syringe pumps (neMESYS, CETONI GmbH, Korbussen, Germany) toward the two inlets of the T-junction geometry. Optical actuation was performed using a laser system (NovaPro 660-125, RGB Lasersystems, Kelheim, Germany) with an emission wavelength of 600 [nm] and a fixed laser input power of 5 [mW]. The laser light was delivered to the device via an SMA connector coupled to a 365 [μm] diameter input optical fiber. The optical detection system used two 365 [μm] diameter output optical fibers connected to two photodiodes, named *PHA* and *PHB* (PDA100A, Thorlabs, Newton, NJ, USA) with a gain of 70 [dB] to detect variations in light intensity. The optical signals were acquired for 120 [s] using a PC oscilloscope (Picoscope 2204A, Pico Technology, Cambridgeshire, UK) operating at a sampling rate of $f_{s}$ = 1.5 [kHz]. Additionally, a color CMOS compact scientific camera (CS165MU, Thorlabs, Newton, NJ, USA) with a resolution of $1440\times 1080\;px^{2}$ (3.45 [μm] square pixels) was used to capture high-resolution images of the slug-flow process at the cross-point of the input microchannels. The camera was connected to a PC via a USB interface for frame acquisition and subsequent analysis.

### 2.4. Experimental Campaign

To investigate immiscible liquid–liquid and gas–liquid slug-flow processes for the MoF device, an experimental campaign was carried out. The *continuous phase* was injected through the main channel, while the *dispersed phase* was supplied through the side channel of the T-junction. For the liquid–liquid slug flow, hexadecane (refractive index $n_{hexadecane}\;=\;1.43$) was used as the *continuous phase* and deionized water ($n_{water}=1.33$) as the *dispersed phase*, whereas for the gas–liquid slug flow, deionized water (refractive index $n_{water}=1.33$) was used as the *continuous phase* and air ($n_{air}=1$) as the *dispersed phase*. Moreover, to maintain the integrity of the process, no surfactants were used in either the liquid–liquid or gas–liquid slug-flow processes, as they could alter key parameters such as slug length or meniscus formation.

For both the liquid–liquid and gas–liquid slug-flow processes, the experimental campaign was divided into three ***subsets:******Subset 1*** ($F_{1}=F_{2}$): The volumetric flow rates for both the *continuous phase* ($F_{1}$) and *dispersed phase* ($F_{2}$) were set to the same value. The flow rates were varied across 10 different values.***Subset 2*** ($F_{1}\neq F_{2}$ and $F_{1}$ = 0.1 [mL/min]): The volumetric flow rate for the *continuous phase* ($F_{1}$) was fixed at 0.1 [mL/min], while the flow rate for the *dispersed phase* ($F_{2}$) was varied across 6 different values.***Subset 3*** ($F_{1}\neq F_{2}$ and $F_{2}$ = 0.1 [mL/min]): The volumetric flow rate for the *dispersed phase* ($F_{2}$) was fixed at 0.1 [mL/min], while the flow rate for the *continuous phase* ($F_{1}$) was varied across 6 different values.

Table 1 provides a summary of the experimental conditions for ***Subset 1***, ***Subset 2***, and ***Subset 3***. It reports the volumetric flow rates in [mL/min] set by the hydrodynamic source, along with the nominal velocity $v_{n}$ in [m/s], calculated as follows:(5)$$v_{n}=\frac{F}{A}$$
where $v_{n}$ is the imposed input flow rate expressed in [m^3^/s] and *A* is the microchannel cross-sectional area in [m^2^].

### 2.5. Acquired Signals and Investigated Responses

Consistent with the working principle described in Section 2.1 for immiscible gas–liquid and liquid–liquid slug flows, the square-wave signals recorded by the two photodiodes exhibited a lower level corresponding to the fluid with the lower refractive index and a higher level associated with the fluid with the higher refractive index.

The acquired signals were post-processed as described below. In detail, a low-pass filter with a cut-off frequency of 200 [Hz] was applied to remove high-frequency harmonics. Next, a smoothing procedure was used to eliminate noise from the signal and reveal the main square-wave pattern. Starting from the two post-processed optical signals, four responses were investigated:The *slug-flow velocity ($v_{slug}$)*—This was estimated using the Dual-Slit Particle Signal Velocimetry (DPSV) method [49] applied to a slug-flow process. Briefly, it relies on the analysis of optical signals recorded by two photodiodes positioned at a known distance *d* in [m] along the flow direction. By cross-correlating the signals, the time delay Δt in [s] between detections is determined through cross-correlation peak extraction, allowing the estimation of the slug-flow velocity according to(6)$$v_{slug}=\frac{d}{\Delta t}.$$The *slug length ($L_{slug}$)*—This was determined by considering the slug-flow velocity $v_{slug}$ in [m/s] (derived from Equation (Equation 6)), the sampling period of the optical signal acquisition system $T_{s}$ in [s], and the number of samples corresponding to a single slug level within the square wave $N_{slug}$. The slug length $L_{slug}$ was then calculated as the product of these three parameters according to(7)$$L_{slug}=v_{slug}\cdot T_{s}\cdot N_{slug}.$$The *Reynolds number (Re)*—This was calculated according to(8)$$Re=\frac{D_{h}\cdot v_{slug}}{\nu_{m}}$$
where $D_{h}$ is the hydraulic diameter of the microchannel in [m]; $v_{slug}$ is the slug velocity in [m/s]; and $\nu_{m}$ is the kinematic viscosity of the fluid in [m^2^/s], evaluated as the weighted average of the kinematic viscosities of the *continuous phase* ($\nu_{1}$) and *dispersed phase* ($\nu_{2}$):$$\nu_{m}=\frac{\nu_{1}+\nu_{2}}{2}$$The *Capillary number (Ca)*—This was calculated according to(9)$$Ca=\frac{\mu \cdot v_{slug}}{s_{t}}$$
where μ is the dynamic viscosity of the *continuous phase* in [Pa ·s], $s_{t}$ is the surface tension between the two fluids in [N·m], and $v_{slug}$ is the slug velocity in [m/s] [56].

For comparison with the analytical results, the investigated responses $L_{slug}$, Re, and Ca were also calculated using the nominal velocity values $v_{n}$ reported in Table 1.

Furthermore, as previously discussed, a subset of the experimental campaign was replicated (refer to Table 2) in the simulation environment using the model outlined in Section 2.2.2. In this instance, the number of experiments was reduced compared to the experimental campaign due to computational time constraints (see Table 3). Specifically, the simulation results were utilized to determine the slug velocity and slug length. To allow for a direct comparison with the experimental data, the same analytical methodology was employed to derive these two parameters, with the key difference being that the phase-field variable (see Section 2.2.1) was used instead of the optical signal. This variable effectively distinguishes the two phases and exhibits similar patterns to those of the optical signal. Consequently, two observation points within the main microchannel were selected at the same position and distance as the optical fibers employed in the MoF device so that the optical signal was reproduced; the same methodology was used.

## 3. Results and Discussion

This section explores and compares both liquid–liquid and gas–liquid slug flows within the MoF device, correlating the experimental results with the theoretical and numerical ones. A microscopy image, obtained during the slug passage for both the liquid–liquid and gas–liquid processes investigated during the experiments, is shown in Figure 3d. Since both hexadecane and water are transparent, a red colorant was added to the water to provide a visual contrast between the two phases. The same methodology was consistently applied to both the experimental data and the simulations, allowing an accurate comparison between the two. The correlation-based time-domain analysis—DPSV—was employed to calculate the slug velocity, which was also used to determine the slug length. Finally, both the experimental and theoretical slug velocities were utilized to determine the Reynolds and Capillary numbers, providing valuable insight into the process and facilitating a comparison between the theoretical estimation and the real conditions within the device. All data analysis and parameter evaluations were performed using MATLAB software (vR2024b).

### 3.1. Liquid–Liquid Slug Flow: Hexadecane–Water

The study of the microfluidic process started by estimating the slug velocity; once this was determined, other characteristic parameters could be derived from it. Regarding the simulation analysis, the slug velocity was evaluated by analyzing the phase-field variable, which in this case was employed to simulate the two optical signals. Specifically, the two observation points $P_{1}$ and $P_{2}$ were positioned at the same locations as the optical fibers to capture their respective patterns, as shown in Figure 3c, illustrating the distribution of the two fluids inside the microchannel. It is clear that the phase-field variable followed a similar trend to that of the optical signals, differing only in that it ranged between −1 and 1, depending on the specific phase present at any given point in the microchannel investigation area. Although the signals exhibited identical waveforms, they were temporally shifted from one another. By employing the known distance between the two points, the time delay between the signals was accurately estimated through cross-correlation analysis, which facilitated the precise calculation of the slug velocity within the microchannel. Figure 5a illustrates the phase-field variable during the slug-flow simulation involving hexadecane–water that was carried out for ***Subset 1*** for an input flow rate of 0.1 [mL/min]. The corresponding cross-correlation functions are depicted in Figure 5b, highlighting the relationships between the two signals. To determine the time delay Δt between them, the peak of the cross-correlation function was used. Given the physical distance between the two points, defined as *d* = 1 [mm], the mean slug velocity was subsequently calculated, as outlined in Equation (Equation 6), providing a comprehensive understanding of the dynamics within the microchannel flow.

The same approach was applied in the experimental analysis to evaluate the slug velocity, enabling a direct comparison between the two sets of results. In this case, the slug velocity was evaluated by analyzing the optical signals captured by two photodiodes, referred to as *PHA* and *PHB*. Although the signals emitted by these photodiodes exhibited similar behavior (the voltage level associated with the two fluids was different from −1 and 1, and depended on the refraction index of the fluid), they were temporally shifted, as before (see Figure 6a). Therefore, for the simulation analysis, by knowing the distance between the two photodiodes, the slug velocity could be determined through the correlation analysis (see Figure 6b). Also, in this case, the peak of maximum correlation, corresponding to the time delay between the two signals Δt, was determined. After that, by applying Equation (Equation 6), the slug velocity was calculated in the same way as for the simulation data. Finally, by extending the analysis to all the experimental campaigns (see Table 1 and Table 2), the slug velocity trend and the characteristic curves for each ***subset*** could be determined.

Figure 7 illustrates the results obtained, organized into the three distinct ***subsets***. The graphs illustrate the relationship between the estimated slug velocity, derived from both the experimental data and simulations, as a function of the nominal velocity defined in Table 1. The black curves represent the theoretical slug velocities, reflecting the ideal behavior of the flow; any deviations from these curves indicate a limitation of the numerical or theoretical model employed. Notably, the data points estimated by the CFD model closely align with the theoretical curves. The deviations can be attributed to the discretization time of the model. Reducing the discretization interval is expected to enhance accuracy, bringing the results closer to the theoretical ideals. This convergence further validates the accuracy of the correlation method utilized for the slug velocity estimation, thereby ensuring consistency with the experimental data. In contrast, the blue curves represent the slug velocities derived from the experimental data. Across all three ***subsets***, the experimental curves are closely aligned with the theoretical curves. In particular, Figure 7a, which depicts the ***subset*** in which the flow rates of the two fluids are equal to each other, shows that the curves are nearly coincident at low slug velocities. However, as the input flow rates increase, the experimental curve begins to deviate upward from the theoretical curve, indicating that the measured slug velocity is greater than that predicted by the theoretical model. A similar trend is observed for the remaining subsets, where the input flow rates are unequal. These results suggest that within the microchannel, there are dynamic phenomena that contribute to an acceleration of the slugs beyond that estimated by the theoretical model.

Moreover, to determine how much the estimated experimental velocities differed from the velocities obtained by the numerical simulation, the percentage error, $e_{\%}$, was computed for each ***subset*** as follows:(10)$$e_{\%}=100\cdot \frac{v_{experimental}-v_{numerical}}{v_{numerical}}$$

When the two input flow rates were equal and perfectly balanced (***Subset 1***), the $e_{\%}$ was very low. Conversely, when the input flow rates were unbalanced (***Subsets 2*** and ***3***), the experimental results began to deviate from the numerical ones, leading to a higher $e_{\%}$, as shown in Figure 7d–f. In addition, the error percentage indicates that the estimated experimental velocity was higher than the numerical one. This suggests that the model does not consider fundamental information such as the presence of the meniscus. This liquid film and its thickness directly affect the slug velocity [57].

Following the determination of the slug velocities, both in the experimental scenario and within the CFD model employed, we investigated the slug length, a critical hydrodynamic parameter that significantly influences inter-slug mass transfer in segmented flow or mixing scenarios [58]. Indeed, both the slug velocity and the slug length between the two phases determine the nature of internal circulation within the slug. Specifically, knowing the slug velocity, the sampling period of the optical signal/phase-field variable $T_{s}$ (see Figure 5a and Figure 6a), and the number of samples corresponding to a single slug level within the square wave $N_{slug}$, it was possible to determine the slug length $L_{slug}$ using Equation (Equation 7). In this case, as with the determination of the slug velocity, the same methodology was applied to both the experimental and CFD models, allowing for a direct comparison between the two. The results are shown in Figure 8 for all three ***subsets*** considered in this work. It was found that the experimental values were in closer agreement with the numerical predictions. The results showed that the slug length ranged from 1 [mm] to 3 [mm] in both the CFD model and the experimental context for both fluids, confirming the validity of the methodology used to determine this parameter. In ***Subset 1***, we observed that the slug length tended to decrease as the input flow rate increased. Initially, the slug lengths of the two phases were distinct from one another; however, as the input flow rate continued to increase, the slug lengths for both phases converged toward the same value. Conversely, when the input flow rates were unequal, as seen in ***Subset 2*** and ***Subset 3***, slight discrepancies arose between the experimental data and the simulation results. The CFD model showed a slug-length inversion occurring almost when the two flows became equal, whereas in the experimental scenario, a clear imbalance in the flows was necessary for the inversion to take place, as illustrated in Figure 8c.

The results show that the CFD model successfully simulated the dynamics of the liquid–liquid slug-flow process with reasonable accuracy. However, as the flow rates increased, the experimental data started to diverge from the theoretical model, underscoring the necessity for in situ monitoring of the process. Additionally, the discrepancies observed in the slug lengths revealed the challenges associated with accurately reproducing the slug formation mechanism. Since this mechanism is crucial to the dynamics of the flow process, it contributed to the differences observed between the numerical and experimental data.

Similar to the previous case, the slug velocities were evaluated by analyzing the phase-field variable with regard to the simulation analysis and the photodiode signals with regard to the experimental analysis. In particular, Figure 9a illustrates the *PHA* signal during the slug-flow experiment involving air–water, carried out in ***Subset 1*** for an input flow rate of 0.1 [mL/min]. The first consideration when comparing the PHA signals of air–water and hexadecane–water slug-flow processes was that the alternation between the two phases was clearly slower. Regarding the liquid–liquid analysis, by knowing the distance between the two photodiodes, the slug velocities could be determined through the correlation analysis (see Figure 9b). Also, in this case, the peak of maximum correlation, corresponding to the time delay between the two signals Δt, was determined. After that, by applying Equation (Equation 6), the slug velocities were calculated for the simulation data. Finally, by extending the analysis to all experimental campaigns (see Table 1 and Table 2), the slug velocity trend and the characteristic curves for each ***subset*** could be determined, as shown in Figure 10.

### 3.2. Gas–Liquid Slug Flow: Air–Water

In this scenario, the results derived from the CFD model were again nearly coincident with the theoretical curves, further confirming the validity of the correlation methodology employed. However, in contrast to the previous case, in which the slug process involved two liquids, the experimental curves in this instance lay below the theoretical curves. This discrepancy indicates that the estimated slug velocities were inadequate to accurately represent this process. The presence of the gas in this flow regime significantly affected the dynamics due to its compressibility, leading to pressure fluctuations within the microchannel. Such compressibility complicated the process estimation and required in situ measurements to fully understand the phenomena within the microchannel.

As in the liquid–liquid case, to determine how much the estimated experimental velocities differed from the velocities obtained by the numerical simulation, the percentage error, $e_{\%}$, was computed for each ***subset*** using Equation (Equation 10). Compared to the previous scenario, the estimated $e_{\%}$ values were much higher, and there were no significant differences across all the considered ***subsets***, as shown in Figure 10d–f. In addition, the $e_{\%}$ values indicated that the estimated experimental velocities were lower than the numerical ones, which is the opposite of the results observed in the hexadecane–water slug flow. In this scenario, the air compressibility and pressure fluctuation, in combination with the presence of the liquid film and leakage flow, directly affected the slug velocities [59].

Following the determination of the slug velocities, both in the experimental scenario and within the CFD model employed, the slug lengths were determined. Therefore, knowing the slug velocities, the sampling periods of the optical signals $T_{s}$, and the number of samples corresponding to a single slug level within the square wave $N_{slug}$, the slug lengths $L_{slug}$ were calculated using Equation (Equation 7), and the results are shown in Figure 11.

Unlike the previous scenario, a noticeable discrepancy of more than one order of magnitude was observed between the experimental results and those derived from the CFD model. In particular, the numerical simulation predicted shorter air slug lengths, whereas the experimental observations showed significantly longer air slugs. Generally, the slug length was influenced by the dynamics of dispersed-phase detachment at the T-junction, where the two phases interact. As discussed in the previous section regarding slug velocities, the compressibility of air played a critical role in these detachment dynamics. This phenomenon may result from increased pressure and flow oscillations caused by air compressibility, which slowed the alternation between the two phases within the microchannel. Such dynamics can delay the breakup of air slugs, a factor not accounted for in the idealized CFD model. Furthermore, pressure disturbances generated by the incoming liquid flow, which dominate the slug breaking mechanism, were not captured numerically. Additionally, it is evident in both scenarios that water dominated the microchannel, with a greater presence relative to air. This can be observed in ***Subset 1***, where water length consistently exceeded air length even with equal input flow rates (see Figure 11a). Similar patterns were observed in the slug-length inversion points in ***Subset 2*** and ***Subset 3***, which only occurred when the air input flow rate exceeded that of water (see Figure 11b,c).

### 3.3. Reynolds and Capillary Numbers

The Reynolds and Capillary numbers were analyzed for both the experimental and nominal cases, similar to the evaluation of the slug lengths. Given the kinematic viscosities of hexadecane ($\nu_{Hex}≃3.97\times 10^{-6}\;\left[m^{2}/s\right]$), water ($\nu_{Wat}≃1.0038\times 10^{-6}\;\left[m^{2}/s\right]$), and air ($\nu_{Air}≃1.5\times 10^{-5}\;\left[m^{2}/s\right]$), along with the hydraulic diameter of the microchannel ($D_{h}=4\times 10^{-4}$ [m]), the Reynolds number can be calculated using Equation (Equation 8), taking into account both the experimental and nominal velocities. Similarly, with the dynamic viscosities of hexadecane ($\mu_{Hex}≃$ 0.003 [Pa·s]), water ($\mu_{Wat}≃$ 0.001 [Pa·s]), and air ($\mu_{Air}≃1.81\times 10^{-5}$ [Pa·s]), as well as the surface tensions between hexadecane and water ($s_{t(HW)}≃0.051$ [N/m]) and air and water ($s_{t(AW)}≃$  0.072 [N/m]), the Capillary number can be determined using Equation (Equation 9), again considering both the experimental and nominal velocities. As discussed in Section 2.2.1, typically, these values are estimated before the experimental phase using the nominal velocities, providing an initial understanding of the process. Indeed, both numbers are critical in explaining the dynamics of the fluidic process.

Based on the values of both the experimental and nominal slug velocities obtained, the estimated Capillary numbers for the hexadecane–water and air–water slug-flow processes were within the range of *Ca* ∈$\left[10^{-4},10^{-2}\right]$, confirming the occurrence of slug formation [60]. Therefore, the predictions regarding slug formation were validated considering the nominal velocities and were correctly predicted. Regarding the Reynolds numbers, the experimental and nominal values for the hexadecane–water slug flow were within the range of *Re* ∈ [1,20], whereas those for the air–water slug flow were within the range of *Re*∈[1,6]. This validates that the laminar-flow regime was accurately estimated. Furthermore, to address the difference between the predictions based on nominal velocities and the real case scenario, the percentage error, $e_{\%}$, for the Reynolds number was computed using Equation (Equation 10).

Considering the hexadecane–water slug flow, the percentage error was within the range $e_{\%}\in \left[0.27,\;26.72\right]$, and for the air–water slug flow, the percentage error was within the range $e_{\%}\in \left[33.04,\;61.6\right]$. These error values highlight the discrepancies in the slug velocities discussed previously, as illustrated in Figure 7. Unlike the numerical values, which accurately predicted both slug formation and flow, the resulting errors indicate that the predictions may be significantly incorrect when considering microfluidic processes near the boundary between different regimes. Specifically, when operating at the limit between laminar and turbulent flow, or at the transition between the slug flow and bubble regimes, the observed errors demonstrate the necessity for experimental validation to confirm the dynamic conditions of the process with certainty.

## 4. Conclusions

This work presents a PDMS-based micro-optofluidic device called an MoF, designed to perform in situ measurements and facilitate a comprehensive understanding of fluid dynamics through a non-invasive optical detection technique. In this paper, we focus on the investigation of slug flow, considering both liquid–liquid (hexadecane–water) and gas–liquid (air–water) processes. An extensive experimental campaign was conducted to validate the approach employed for investigating slug-flow velocity using the Dual-Slit Particle Signal Velocimetry (DPSV) method. The limitations and advantages of the numerical predictions compared to the experimental observations in characterizing slug formation and laminar-flow regimes were investigated. Notable discrepancies between the predicted and actual process conditions, particularly in gas–liquid flows like the air–water slug regimes examined, underscore the limitations of purely theoretical models and estimations based solely on the externally imposed input flow rates. The results highlight the need for in situ and continuous monitoring, particularly for processes operating under critical conditions of instability. Future work will aim to apply this approach to study complex multiphase flow phenomena, particularly meniscus dynamics, contact angle behavior, and other interfacial properties influencing microscale fluid interactions.

## Figures and Tables

**Figure 1 micromachines-16-00586-f001:**
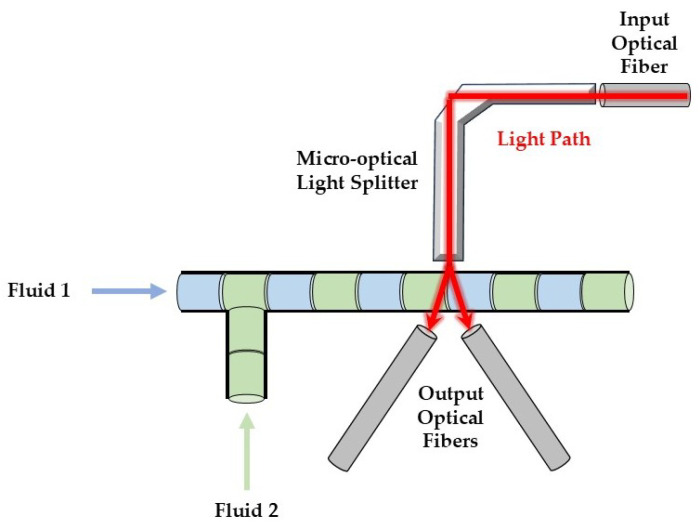
Working principle of the MoF device for immiscible gas–liquid and liquid–liquid slug-flow detection.

**Figure 2 micromachines-16-00586-f002:**
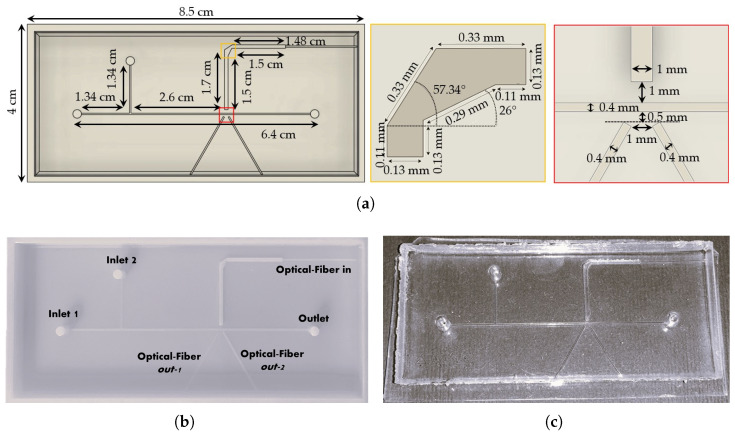
(**a**) Top view of the CAD design for the MoF device. The yellow box highlights the dimensions of the micro-optical light splitter, and the red box shows a close-up view of the optical investigation area with its dimensions. (**b**) 3D-printed master mold of the MoF device. (**c**) Final assembled PDMS MoF device.

**Figure 3 micromachines-16-00586-f003:**
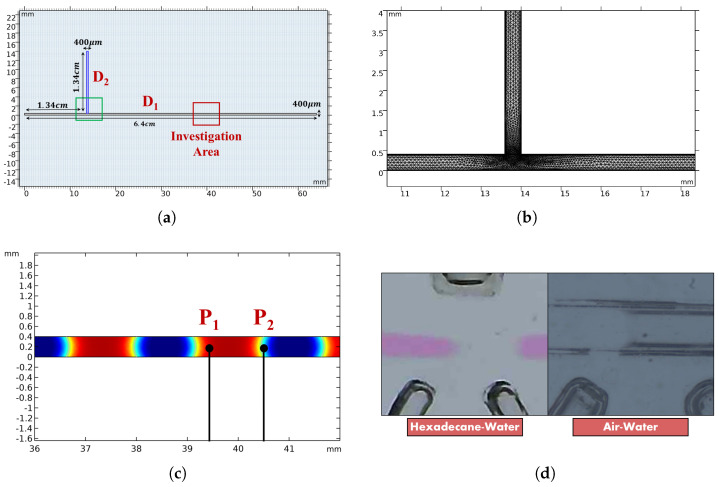
(**a**) Micro-optofluidic device geometry. (**b**) Close-up view of the mesh distribution involved in the CFD simulation environment at the T-junction. (**c**) Locations of $P_{1}$ and $P_{2}$, representing the observation points in the model used to simulate the presence of the two photodiodes. (**d**) Investigation areas of both the hexadecane–water and air–water processes.

**Figure 4 micromachines-16-00586-f004:**
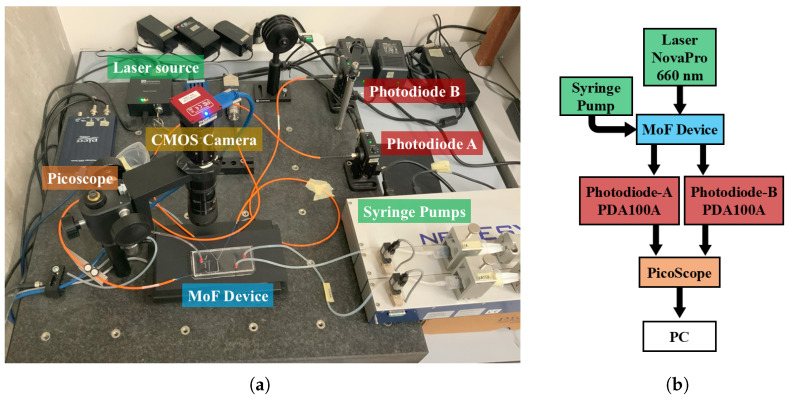
(**a**) Experimental setup. (**b**) Block diagram of the system employed during the experimental campaign.

**Figure 5 micromachines-16-00586-f005:**
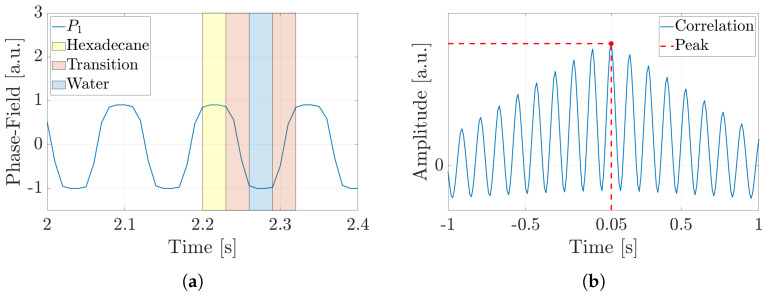
Numerical results of CFD simulations. (**a**) Hexadecane slug presence—high level highlighted in yellow; water slug passage—low level depicted in blue; and transition between these two events, displayed in red. (**b**) Cross-correlation between the phase-field variables $P_{1}$ and $P_{2}$ to compute their time delay.

**Figure 6 micromachines-16-00586-f006:**
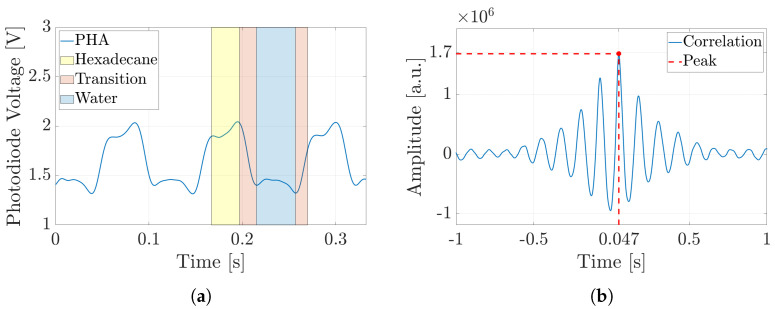
Experimental result trends of the optical signals acquired by *PHA* (Photodiode-A) for hexadecane–water slug flow in ***Subset 1***, with an input flow rate of 0.1 [mL/min]. (**a**) Hexadecane slug presence—high level highlighted in yellow; water slug passage—low level depicted in blue; and transition between these two events, displayed in red. (**b**) Cross-correlation between the optical signals of PHA and PHB (Photodiode-A and Photodiode-B) to compute their time delay.

**Figure 7 micromachines-16-00586-f007:**
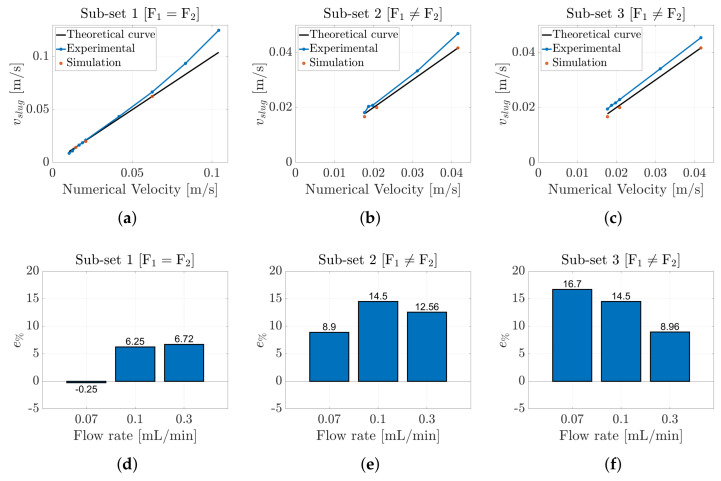
(**a**–**c**) Comparison between the theoretical, numerical, and experimental slug velocities for hexadecane–water slug flow for the three different ***subsets*** reported in Table 1 and Table 2. Percentage error $e_{\%}$ computed between the numerical and experimental results for ***Subset 1*** (**d**), ***Subset 2*** (**e**), and ***Subset 3*** (**f**).

**Figure 8 micromachines-16-00586-f008:**
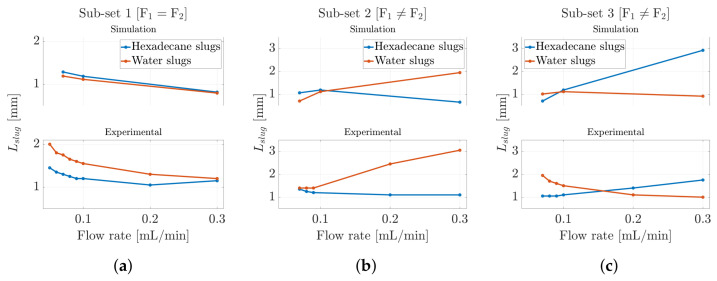
Slug length trend for hexadecane–water slug flow for (**a**) ***Subset-1***, (**b**) ***Subset-2***, and (**c**) ***Subset-3***, as reported in Table 1 and Table 2.

**Figure 9 micromachines-16-00586-f009:**
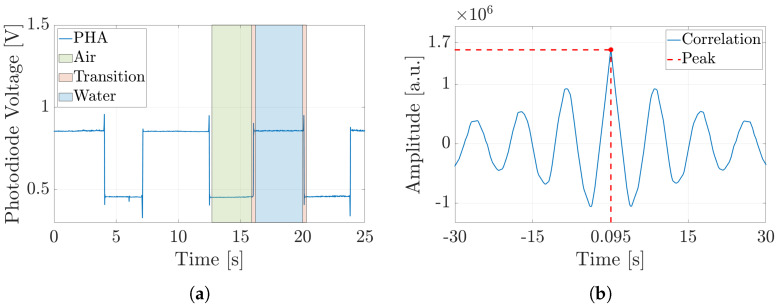
Experimental result trends of the optical signals acquired by *PHA* for air–water slug flow in ***Subset 1***, with an input flow rate of 0.1 [mL/min]. (**a**) Air slug presence—low level highlighted in green; water slug passage—high level depicted in blue; and transition between these two events, displayed in red. (**b**) Cross-correlation between the optical signals of PHA and PHB to compute their time delay.

**Figure 10 micromachines-16-00586-f010:**
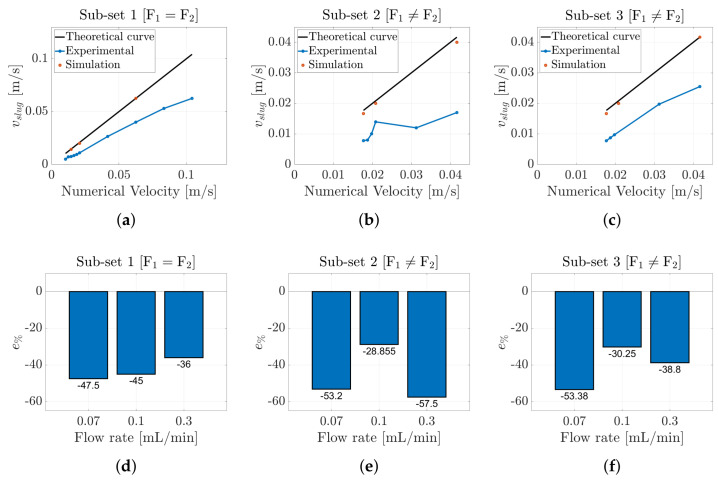
(**a**–**c**) Comparison between the theoretical, numerical, and experimental slug velocities for air–water slug flow for the three different ***subsets*** reported in Table 1 and Table 2. Percentage error, $e_{\%}$, computed between the numerical and experimental results for ***Subset 1*** (**d**), ***Subset 2*** (**e**), and ***Subset 3*** (**f**).

**Figure 11 micromachines-16-00586-f011:**
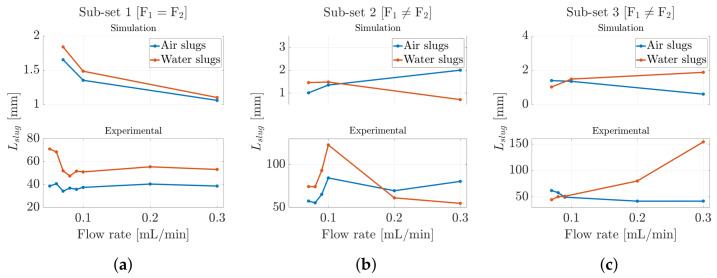
Slug-length trend for air–water slug flow for (**a**) ***Subset-1***, (**b**) ***Subset-2***, and (**c**) ***Subset-3***, as reported in Table 1 and Table 2.

**Table 1 micromachines-16-00586-t001:** Experimental campaign highlighting the experimental conditions for ***Subset 1***, ***Subset 2***, and ***Subset 3***. The table reports the volumetric flow rates in [mL/min] set by the hydrodynamic source, along with the nominal velocities $v_{n}$ in [m/s].

*Subset 1*		*Subset 2*		*Subset 3*	
($F_{1}=F_{2}$)		($F_{1}\neq F_{2}$)		($F_{1}\neq F_{2}$)	
		($F_{1}$ = 0.1 [mL/min])		($F_{2}$ = 0.1 [mL/min])	
($F_{1}$, $F_{2}$)		($F_{2}$)		($F_{1}$)	
[mL/min]	[m/s]	[mL/min]	[m/s]	[mL/min]	[m/s]
0.05	0.0052				
0.06	0.0062				
0.07	0.0073	0.07	0.0089	0.07	0.0089
0.08	0.0083	0.08	0.0094	0.08	0.0094
0.09	0.0094	0.09	0.0099	0.09	0.0099
0.1	0.01	0.1	0.01	0.1	0.01
0.2	0.021	0.2	0.015	0.2	0.015
0.3	0.031	0.3	0.021	0.3	0.021
0.4	0.042				
0.5	0.052				

**Table 2 micromachines-16-00586-t002:** Experimental conditions reproduced in the simulation environment.

*Subset 1* [mL/min]	*Subset 2* [mL/min]	*Subset 3* [mL/min]
($F_{1}=F_{2}$)	($F_{1}\neq F_{2}$)	($F_{1}\neq F_{2}$)
	($F_{1}=0.1$)	($F_{2}=0.1$)
($F_{1}$, $F_{2}$)	($F_{2}$)	($F_{1}$)
0.07	0.07	0.07
0.1	0.1	0.1
0.3	0.3	0.3

**Table 3 micromachines-16-00586-t003:** Simulation details regarding the computational time, mesh dimensions, and time step for ***Subset 1***, ***Subset 2***, and ***Subset 3***.

		Computational Time [hh:mm:ss]	Minimum Mesh Size Length [μm]	Maximum Mesh Size Length [μm]	Time Step [ms]
* **Subset 1** *	0.07[mL/min]	16:12:17	4	140	10
	0.1[mL/min]	22:45:52	4	140	10
	0.3[mL/min]	26:21:56	4	140	1
* **Subset 2** *	0.07[mL/min]	43:30:34	1.6	112	10
	0.1[mL/min]	22:45:52	4	140	10
	0.3[mL/min]	47:35:48	1.6	112	5
* **Subset 3** *	0.07[mL/min]	39:05:10	1.6	112	10
	0.1[mL/min]	22:45:52	4	140	10
	0.3[mL/min]	48:25:34	1.6	112	5

## Data Availability

The raw data supporting the conclusions of this article will be made available by the authors on request. The original contributions presented in the study are included in the article/Appendix A, further inquiries can be directed to the corresponding author.

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
