# Peer review of "Experimental and Numerical Study of Slug-Flow Velocity Inside Microchannels Through In Situ Optical Monitoring"

_micromachines, 2025, doi:10.3390/mi16050586_

Round 1
Reviewer 1 Report
Comments and Suggestions for Authors
The manuscript "Experimental and Numerical Study of Slug-Flow Velocity inside Microchannels by in-situ Optical Monitoring" focuses on an important field of microfluidics, which is combination of microfluidic confinement with optical analytical methods. Optofluidics offers non-invasive tools for exploring microscale processes in lab-on-chip devices.
In this paper, the authors discuss optofluidic approaches to evaluating flow velocities in slug-flow modes and monitoring of microchannel dynamics.
The topic of the manuscript fits the scope of Micromachines and the Special Issue "Complex Fluids Flows in Microchannels".
The manuscript produces an impression of a solid work with thorough and detailed experimental and numerical approaches. However, its review reveals certain issues and concerns, which should be addressed in the revision so the submission could be further considered for publication.
- A combination of optical fibers and microfluidic channels is a well-known approach. To emphasize the novelty of their manuscript, the authors are recommended to enrich their literature review in the Introduction by relevant references to compare their approach to optofluidic chip design with competitive microfluidic/optofluidic techniques for flow characterization.
- The Introduction is rather bulky and hard to read. The aim of the study is described in Lines 59-64. Then, additional details of the experimental design (Lines 65-80) and review of literature (Lines 81-106) are provided. The authors are recommended to revise the literature review by stating the aim of the study and its major outcomes in the end of Introduction and moving experimental details to the respective section of the manuscript.
- The device was manufactured by 3D printing using liquid photopolymer. What is the accuracy of this method? A major concern is the resulting surface roughness of side walls and its impact of flow dynamics. The authors are recommended to add a microscopy image of a microfluidic channel to Fig. 2 with a magnification sufficient to evaluate surface roughness.
- The width of the main channel shown in Fig. 2 is 400 μm. Can 3D printing be used for fabricating smaller channels with the width of 50-100 μm? Such a range of widths is typical for microfluidic devices. Compatibility of the proposed flow analysis techniques with narrower microchannels is an important aspect of application potential of such optofluidic chip designs.
- In droplet microfluidics, surfactants are commonly used to provide a smooth and reliable generation of microdroplets. However, no information about surfactant was found in the manuscript. If no surfactant was used, how did it affect stability of slug flow generation? Please, comment.
- The manuscript will benefit from adding a microscopy image of microfluidic slug flow for liquid-liquid and liquid-air systems observed during experiments.
- Why was the 5:1 ratio of the silicon elastomer and the curing agent used for the bulk cover layer? A standard ratio is 10:1. How did it influence the resulting cured PDMS characteristics?
- How do the velocities reported in Fig. 7 and 10 relate to the velocity distribution across the microchannel?
Minor comments:
- In the Experimental, information about software used for numerical simulations and optical data analysis should be added.
- The English is fine, however, the authors should check their manuscript to eliminate undesirable typos such as those in Lines 1 and 116.
- In Line 116, it looks like that the text was not finalized by the authors.
- Provide appropriate references to Eq. 9 (Line 316) and confirmation of the slug flow mode according to the Ca values (Line 491).
Author Response
Please see the attachment.
Thanks in advance.
Best regards,
Samuele Moscato, Emanuela Cutuli, Massimo Camarda and Maide Bucolo

Reviewer 2 Report
Comments and Suggestions for Authors
The authors demonstrate a PDMS-based microoptofluidic device that provides direct optical access for flow velocity estimation within microchannels, eliminating the need for bulky components and external instrumentation tools. However, several flaws and errors need to be rectified. Also, the novelty of the proposed work is unclear. Below are the comments:
1) Page 1, line 1, "reliable" should be changed to reliable
2) Keywords should be restricted to five
3) The abstract should highlight the key statistical information
4) The introduction should be comprehensively discussed, including the fundamentals of microfluidics, optics, and fluid dynamics within microcapillary
5) Authors can consider adding these relevant references to support their work: 10.1016/j.bej.2023.109027; 10.3390/en15197284
6) Add a statistical comparison table between simulation and experimental data.
7) In figure 2(b) & (c), please label the dimensions, also name the inlet and outlet
8) Figure 4(a) increase the font size in the blocks mentioned
9) Figure 4(b) can be removed or reordered because it looks so messy with all the component arrangements
10) Add the swot analysis for experimental vs simulation results
11) What are the limitations of simulation and its advantages over experimental
12) Future directions in microfluidic technology
13) References should be rechecked, and more recent, relevant references added
14) The English language should be rechecked, and grammatical errors should be rectified.
Author Response

(The authors gave the same response as above.)

Reviewer 3 Report
Comments and Suggestions for Authors
Review of the paper
Experimental and Numerical Study of Slug-Flow Velocity inside Microchannels by in-situ Optical Monitoring
1) Please check English throughout the paper. E.g. Abstract ” reliabl”.(Line 1).
Line 175 “Kg/m3”.
2) Line 116 “…(Section ??) and slug length (Section ??)”. Please correct this
3) Please provide the objective of this work in an explicit form, at the end of the Introduction section.
4) Fig. 1 and related text. The principles of the device are not clear enough. How does micro-optical light splitter work? Why the splitting takes place?
Line 127 “with three optical fiber insertions”. Where are these optical fiber insertions?
Line 129 “directed to two close portions”. How close are these portions.
In other words, this part should be extended providing more details.
5) A few photographs of the used MoF device (separately and installed to the microchannel) should be added.
6) Fig. 2a. At the output there are two output optical fibers. Why there is an angle between them? How large is this angle?
7) Fig. 2. There are sizes with precision of up to 0.01 mm (0.13 mm, 0.33 mm, etc.). PDMS is a quite soft material. How these sizes have been checked instrumentally?
8) Line 177. “μ is the viscosity coefficient expressed in [m2/s]”. In fact it has “Pa s” units.
9) Line 186 “where the density and viscosity of each fluid…”. Please specify which fluid has index “1” and which one has “2” in eqs (3) and (4).
10) Tables 1 and 2. Specify in each column the name of variable listed (F1, F2 or both).
11) Fig. 6. What mean “PHA” and “PHB”? This should be explained in the figure caption.
12) Fig. 10. There is a very large deviation between theoretical and experimental lines. The same concerns are related to the Fig. 11.
I guess these is a great disadvantage of the used method, and they should be corrected before the authors resubmit their paper to the journal.
Author Response

(The authors gave the same response as above.)

Reviewer 4 Report
Comments and Suggestions for Authors
The authors studied the slug-flow of two immiscible fluids in a microchannel through an extensive experimental campaign to evaluate flow velocity in both liquid-liquid and gas-liquid regimes.
- The present abstract needs to be rewritten. The present form could not be the focus of a proper abstract. Authors should briefly clarify the background, research gap, and objective of the present study at the beginning.
- Is the last para of introduction section necessary? It just explains about structure of paper (all sections).
- The authors said, “unstructured triangular mesh are used”. However, there is no clear details of grid independency of results. It is very essential for numerical simulation. Please include the grid independency test.
- In fluid dynamics, kinematic viscosity is usually denoted by the symbol “ν” (Greek letter \nu). Why do authors used different symbol “σ” (\sigma), which is usually used to denote Electrical conductivity of fluid. It is better to use standard symbols to avoid the confusion.
- It is better to present the clear Research Question(s)? What is already known? What is unknown? Why the study is important?
- Numerical method is not clear. Which method is used to solve the model in CFD?
- What scheme is used for convective and diffusive terms in Navier-Stokes equations in CFD?
- What is the convergence condition of the numerical simulation?
- Did you make your own code for numerical simulation or use any commercial software? Give the details.
- The conclusion should be brief and concise.
Author Response

(The authors gave the same response as above.)

Round 2
Reviewer 1 Report
Comments and Suggestions for Authors
The authors addressed my comments and concerns in their detailed revision.
The manuscript will be a good contribution to the field.
Reviewer 2 Report
Comments and Suggestions for Authors
Authors have addressed the comments raised and now it can be accepted in its current form
Reviewer 3 Report
Comments and Suggestions for Authors
The authors have thoroughly improved the paper.
It could be recommended now for publication.
Reviewer 4 Report
Comments and Suggestions for Authors
Please, Add the version of the COMSOL Multi-Physics software.